

# Bullying victimization and child sexual abuse among left-behind and non-left-behind children in China

Li Yan[1], Qianqian Zhu[2], Xiaowen Tu[1], Xiayun Zuo[1], Chunyan Yu[1,3], Chaohua Lou[1] and Qiguo Lian[1,3]

[1] Key Lab of Reproduction Regulation of NPFPC, SIPPR, IRD, Fudan University, Shanghai, China
[2] Shanghai Ninth People's Hospital, Shanghai Jiaotong University School of Medicine, Shanghai, China
[3] School of Public Health, Fudan University, Shanghai, China

Corresponding author
Qiguo Lian, qglian@fudan.edu.cn

## ABSTRACT

**Background**. Bullying is one of the most important factors associated with child abuse. However, robust tests supporting the assumption that being bullied can contribute to child sexual abuse (CSA) among left-behind children (LBC) remain sparse. This study aims to investigate the association of bullying victimization with CSA among LBC in China.

**Methods**. A cross-sectional study was conducted in six middle schools of Sichuan and Anhui province in 2015. The bullying victimization was assessed by seven items from the Revised Olweus Bully/Victim Questionnaire. The experience of CSA was measured by ten items CSA scale with good consistency.

**Results**. A total of 1,030 children met the sampling criteria, including 284 LBC and 746 non-LBC. The prevalence of CSA was 22.89% in LBC and 20.19% in non-LBC ($p > 0.05$). Bullying victimization was related to CSA among both LBC (adjusted Odds Ratio [aOR] = 2.52, 95% CI [1.34–4.73]) and non-LBC (aOR = 2.35, 95% CI [1.58–3.53]). The association between bullying victimization and CSA was much higher among left-behind girls (left-behind girls: aOR = 7.36, 95% CI [2.16–24.99]; non-left-behind girls: aOR = 2.38, 95% CI [1.08–5.27]). Also, LBC of a young age (11–15), children with siblings, living in rural areas and non-traditional family structure who were bullied were more likely to suffer CSA than their non-LBC peers.

**Conclusions**. Bullying victimization is associated with a significant increase in CSA among both LBC and non-LBC. Anti-bullying programs should target vulnerable populations including female LBC and LBC with siblings to reduce the risk of CSA.

# INTRODUCTION

Child sexual abuse (CSA) can occur regardless of culture, ethnic heritage, gender or income level, and the prevalence of CSA ranged from 8% to 31% for girls and 3% to 17% for boys around the whole world (*Barth et al., 2013*). In China, the prevalence of CSA experience before the age of 16 years ranged from 10.2% to 35.2%, showing that CSA has become a huge concern in China (*Yu et al., 2017*). CSA includes a broad and often inconsistent range of behaviors in different studies. Some studies include only physical contact, whereas

others also include non-contact sexual abuse. The World Health Organization defines sexual abuse as "the involvement of a child in sexual activity that he or she does not fully comprehend, is unable to give informed consent to, or for which the child is not developmentally prepared, or else that violates the laws or social taboos of society" (*WHO, 2010*). CSA has a negative impact on the psychological and educational functions of adolescents and youth (*Meinck, Cluver & Boyes, 2015a*). The variety of adverse outcomes include risks for bullying victimization (*Hebert et al., 2016*), unintended pregnancy and sexual abuse perpetration (*Meinck, Cluver & Boyes, 2015a*). Exiting research has identified some individual risk factors that increase risks for CSA, including gender, age, family constellation, parental characteristics, roles of school bullying, and so on (*Meinck et al., 2015b*).

School bullying, including physical, verbal, property and relational bullying, is pervasive around the world (*Coyne, Archer & Eslea, 2006*). In China, 66.1% of boys and 48.8% of girls suffered one or more kinds of school bullying; 8.1% of boys and 2.9% of girls experienced more than four types of school bullying in 2009 (*Yi-juan et al., 2009*). The newly reported incidences of bullying victimization, perpetration and witnessing were 26.10%, 9.03%, and 28.90% respectively, according to the updated national representative survey data collected in 2016 (*Han, Zhang & Zhang, 2017*). Most importantly, the consequences of school bullying can be severe and persistent, including mental health problems, self-harm, and suicide (*Lereya et al., 2015*). Some youths bullied in school may also be the victims in other settings, such as in their family and communities (*Holt, Finkelhor & Kantor, 2007*). A study showed that physical assault in the community doubled the risk of sexual abuse for girls (*Meinck, Cluver & Boyes, 2015a*). Another study found that school bullying was associated with increased risk of CSA, the prevalence of CSA was 12.1% in victims and decreased to 3.1% in non-involved students (*Holt, Finkelhor & Kantor, 2007*).

The left-behind children (LBC), a particular group with a higher risk for mental health problems in rural China, are defined as the children who stay in rural areas more than six months and have one or both their parents heading to urban areas for work (*Wang et al., 2015*). The number of LBC in China has reached 61.02 million in 2010, accounting for 21.88% of all children in China (*Liu et al., 2016*). Although millions of migrant parents pursue better opportunities for their children, their children left behind actually live in a situation lacking parental care and nurturance and are more likely to suffer abuse than others (*Zhao, Liu & Wang, 2015*). LBC were more likely to report being bullied (*Otake, Liu & Luo, 2017*), and suicide attempts (*Chang et al., 2017*) than non-LBC did in China.

That children bullied by their peers are more likely to suffer sexual abuse can be explained somewhat by the developmental victimology framework (*Finkelhor & Dziuba-Leatherman, 1994*). Developmental victimology is the study of the broad spectrum of childrens' victimizations during all childhood, and the effort to understand the overlaps, common risk factors, interrelationships, and sequencings. The explanations show that children who are involved in any kind of victimization such as family instability, lack of supervision, and bullying might create a risk for additional types of victimization (*Finkelhor & Dziuba-Leatherman, 1994*).

Although studies examined risk factors associated with school bullying and sexual abuse separately (*Hinduja & Patchin, 2012*; *Madu & Peltzer, 2001*), there is scientific evidence indicating that bullying victimization was a potential risk factor for CSA (Hebert et al. 2016). The survey including 8194 students aged 14–18 years attending public and private schools in Quebec showed that school bullying was significantly associated with CSA ($\beta = 0.36$; $p < 0.001$) as well as cyberbullying ($\beta = 0.40$; $p < 0.001$). However, the link between CSA and bullying victimization remains unexplored mainly among LBC in China. Hence, there is an urgent need to understand the importance of bullying victimization as the predictor for CAS among LBC and non-LBC in China. Also, it is essential to identify changeable factors that may amplify the risk. We hypothesized that (1) bullying victimization was positively associated with experience of CSA, and (2) the association in LBC was stronger than that in non-LBC.

## METHOD

### Study site and participants

The study was a cross-sectional study conducted in Anhui and Sichuan provinces from January 2015 to May 2015. Those two provinces are the top two most prominent origins of migrants in China (*Liu et al., 2015*; *Yang et al., 2016*). The target population comprised LBC and non-LBC aged 11–18 years in middle schools and high schools. In this study, we defined LBC as children who were left behind by either one or both parents and stayed at home with extended family members more than six months (*Fan et al., 2010*; *Wang et al., 2015*).

We selected one junior high school and one senior high school in Sichuan province and two junior high schools and two senior high schools in Anhui Province, using purposive sampling to oversample the LBC. We adopted random cluster sampling to select 2–3 classes in grades 7–12 in each school. All students in each selected class were included in the survey. We recruited 1,063 children and dropped 33 children aged over 18, resulting in a sample size of 1,030.

The study was reviewed and approved by the institutional review board of Shanghai Institute of Planned Parenthood Research (PJ2015-05). All participants provided oral consent before the survey. Passive consents were obtained from school administrators and children's parents/guardians.

### Measures

In this study, a questionnaire survey was conducted using computer-assisted self-interviewing (CASI) technology in school computer labs. The content of the questionnaire included the background information (individual characteristics, family, and school information), the experiences of bullying victimization and CSA.

### Bullying victimization

According to the Revised Olweus Bully/Victim Questionnaire by Dan Olweus which has been used in China (*Tang et al., 2018*), we selected seven items to identify the victims of bullying. The internal consistency of the scale was acceptable (Cronbach's alpha = 0.79). The seven items comprising the scale are listed as follows:

a. I was called mean names, was made fun of, or teased in a hurtful way.
b. Other students left me out of things on purpose, excluded me from their group of friends, or completely ignored me.
c. I was hit, kicked, pushed, shoved around, or locked indoors.
d. Other students told lies or spread false rumors about me and tried to make others dislike me.
e. Other students made sexual jokes, comments, or gestures to me.
f. I was bullied by my classmates using a computer or e-mail message or pictures.
g. I was threatened or hurt by my classmates with swords, clubs and other weapons.

The responses were coded as "0 =never, $1 = 1 - 2$ times/year, $2 = 3 - 5$ times/year, $3 = 6 - 11$ times/year, $4 = 1 - 2$ times/month, $5 = 1 - 2$ times/week, $6 \geq 2$ times/week".

The total score ranged from 0 to 42, with higher scores indicating severer experiences of being bullied. We divided the children into two groups (low and high score victims) according to the median score due to non-normal distribution.

## CSA

We used CSA scale with ten items to measure the CSA experience (*Chen, Dunne & Wang, 2002*), and the internal consistency was also acceptable (Cronbach's alpha = 0.69). The CSA scale includes two forms of CSA (non-physical contact and physical contact) and the ten items are listed below (*Halperin et al., 1996*):

a. Has an adult or an older child ever not respected you by demanding you or forcing you to look at pornographic pictures, drawings, films, videotapes or magazines?
b. Has an adult or an older child ever not respected you by sexually explicit talk or hostile language?
c. Has an adult or an older child ever not respected you by demanding you or forcing you to be naked and to expose your genitals for picture taking or filming?
d. Has an adult or an older child ever not respected you by demanding you or forcing you to look at his/her genitals or watch him/her masturbate?
e. Has an adult or an older child ever peeked at your breast or genital?
f. Has an adult or an older child ever not respected you by demanding you or forcing you to be fondled (caresses, rubs, kisses on the whole body and/or your genitals)?
g. Has an adult or an older child ever not respected you by demanding you or forcing you to be fondled him/her (caresses, rubs, kisses on the whole body and/or his/her genitals)?
h. Has an adult or an older child ever not respected you by demanding you or forcing you to submit to having his/her fingers or an object introduced into your body?
i. Has an adult or an older child ever tried to making anal sex or vaginal sex with you?
j. Has an adult or an older child ever made anal sex or vaginal sex with you?

The options of the ten items were all binary (1 = yes, 0 = no).The children suffered CSA if they answered yes to any of these items (*Hawton et al., 2018*; *Peterson et al., 2018*; *Sanchez et al., 2017*).

## Covariates

We controlled eight variables (age, gender, only-child, current place of residence, family structure, relationship with mother and father, parental educational level) as potential confounders because they were associated with the CSA and school bullying (*Wolke et al., 2013*).

We labeled children as only-child if they had no siblings. We recoded the current place of residence as urban area and rural area. We defined family structure as traditional if students' biologic parents married to each other at the time of the survey. We assessed parent–child relationship using the questions "how is your relationship with your mother and father respectively" with options "1 = good, 2 = general, and 3 = poor". We measured parental educational level by the higher level of educational attainment of both parents, ranging from 1 = below junior high school to 3 = college degree or above.

## Statistical analysis

The Stata/SE 15.1 (StataCorp, LLC, College Station, TX, USA) was used to analyze the data. Descriptive analyses were performed using frequencies and percentages. Chi-square tests were performed to evaluate differences in categorical variables including socio-demographic characteristics among LBC and non-LBC. The median was used to divide the bullying scores into two groups due to skewed distribution.

We investigated the association between bullying victimization and CSA using logistic regression models, with and without controlling for potential confounders, including age, only-child, current place of residence, family structure, relationship with mother and father and parental educational level among LBC and non-LBC. Also, subgroup analysis stratified by gender, age, only child, current place of residence and family structure were performed. The statistical significance was considered at $p < 0.05$ for two-sided tests.

# RESULTS

## General information

The final sample consisted of 1,030 children, including 284 LBC (27.57%) and 746 non-LBC (72.43%). Among the 284 LBC, the prevalence of being left behind by their father only, mother only or both parents was 43.31%, 13.73% and 42.96% respectively.

The demographic characteristics of the LBC and non-LBC groups were shown in Table 1. We didn't observe any significant differences in age, gender, only child and parental educational level between the two groups.

A total of 76.76% of LBC lived in rural areas, and the rate was 65.95% in non-LBC. The rate of living with a divorced or single parent was 18.66% in LBC and 11.26% in non-LBC. The two differences were both statistically significant ($p < 0.05$).

The rate of poor relationship with mother was 4.24% and 2.43% in LBC and non-LBC, the rate of poor relationship with father was 6.03% and 3.52% in LBC and non-LBC. These differences were also both significant ($p < 0.05$), indicating that LBC tended to have a poor relationship with parents than their peers. The rate of bullying victimization was similar in LBC and non-LBC, and so did CSA (Table 1).

**Table 1 Characteristics of the study population ($n = 1,030$).**

| Variables | Total ($n = 1,030$) | LBC ($n = 284$) | Non-LBC ($n = 746$) |
|---|---|---|---|
| **Age (years)** | | | |
| 11–15 | 497(48.25) | 135(47.54) | 362(48.53) |
| 16–18 | 533(51.75) | 149(52.46) | 384(51.47) |
| **Gender** | | | |
| Boys | 485(47.09) | 127(44.72) | 358(47.99) |
| Girls | 545(52.91) | 157(55.28) | 388(52.01) |
| **Grade** | | | |
| Junior high school | 489(47.48) | 134(47.18) | 355(47.59) |
| High school | 541(52.52) | 150(52.82) | 391(52.41) |
| **Only child** | | | |
| Yes | 481(46.70) | 144(50.70) | 337(45.17) |
| No | 549(53.30) | 140(49.30) | 409(54.83) |
| **Home place** | | | |
| Urban | 320(31.07)[#] | 66(23.24) | 254(34.05) |
| Rural | 710(68.93) | 218(76.76) | 492(65.95) |
| **Family structure** | | | |
| Traditional | 893(86.70)[*] | 231(81.34) | 662(88.74) |
| Non-traditional | 137(13.30) | 53(18.66) | 84(11.26) |
| **Relationship with mother** | | | |
| Good | 899(88.05)[#] | 231(82.21) | 668(90.27) |
| General | 92(9.01) | 38(13.52) | 54(7.30) |
| Poor | 30(2.94) | 12(4.27) | 18(2.43) |
| **Relationship with father** | | | |
| Good | 832(81.49)[*] | 217(76.95) | 615(83.22) |
| General | 146(14.30) | 48(17.02) | 98(13.26) |
| Poor | 43(4.21) | 17(6.03) | 26(3.52) |
| **Parental education level** | | | |
| Low | 796(78.35) | 226(80.43) | 570(77.55) |
| General | 167(16.44) | 48(17.08) | 119(16.19) |
| High | 53(5.22) | 7(2.49) | 46(6.26) |
| **Bullying victimization score** | | | |
| Low | 624(60.58) | 161(56.69) | 463(62.06) |
| High | 406(39.42) | 123(43.31) | 283(37.94) |
| **CSA victims** | | | |
| No | 822(79.81) | 219(77.11) | 603(80.83) |
| Yes | 208(20.19) | 65(22.89) | 143(19.17) |

**Notes.**
[*]$p < 0.05$.
[#]$p < 0.01$.

**Table 2  Crude associations between bullying victimization and CSA, stratified by gender, age, only child, home place, family structure.**

| | Total OR (95% CI, *p* value) | LBC OR (95% CI, *p* value) | Non-LBC OR (95% CI, *p* value) |
|---|---|---|---|
| Bullying victimization | 2.20(1.62–3.00, <0.001) | 2.04(1.17–3.58, 0.013) | 2.25(1.55–3.25, <0.001) |
| Gender | | | |
| Boys | 1.76(1.20–2.59,0.004) | 1.33(0.64–2.76,0.439) | 1.97(1.25–3.09,0.003) |
| Girls | 3.23(1.77–5.90, <0.001) | 6.01(1.90–19.09,0.002) | 2.30(1.10–4.82,0.027) |
| Age (years) | | | |
| 11–15 | 2.78(1.72–4.47, <0.001) | 2.56(1.03–6.38,0.044) | 2.86(1.63–5.00, <0.001) |
| 16–18 | 2.03(1.32–3.10,0.001) | 2.04(0.97–4.29,0.062) | 1.97(1.17–3.32,0.011) |
| Only child | | | |
| Yes | 1.94(1.25–3.01,0.003) | 1.57(0.73–3.38,0.252) | 2.16(1.26–3.69,0.005) |
| No | 2.48(1.60–3.82, <0.001) | 2.89(1.24–6.75,0.014) | 2.31(1.39–3.85,0.001) |
| Home place | | | |
| Urban | 1.52(0.89–2.61,0.129) | 0.91(0.29–2.84,0.875) | 1.75(0.94–3.23,0.076) |
| Rural | 2.64(1.81–3.85, <0.001) | 2.66(1.38–5.11,0.003) | 2.60(1.63–4.13, <0.001) |
| Family structure | | | |
| Traditional | 2.11(1.52–2.93, <0.001) | 2.05(1.11–3.75,0.020) | 2.12(1.44–3.13, <0.001) |
| Non-traditional | 3.72(1.36–10.21,0.011) | 3.14(0.57–17.23,0.189) | 4.23(1.20–14.87,0.024) |

## Association between bullying victimization and CSA

The results of the crude odds ratio (cOR), adjusted odds ratio (aOR) and their 95% confidence intervals (CIs) were reported in Tables 2 and 3. Bullying victimization was significantly related to CSA (cOR = 2.04 and aOR = 2.52 for LBC, cOR = 2.25 and aOR = 2.35 for non-LBC), suggesting that victims of bullying were more likely to suffer sexual abuse among both LBC and non-LBC.

The results also indicated that victimization was significantly associated with higher proportion of CSA among female LBC (cOR = 6.01 and aOR = 7.36) than that of female non-LBC (cOR = 2.30 and aOR = 2.38). Participants who were bullied were at higher risk of being sexually abused in younger age (cOR = 2.56 and aOR = 3.42 for LBC, cOR = 2.86 and aOR = 3.32 for non-LBC) than those aged 16–18 years. Non-traditional family structure puts the bullying victims at higher chance of being sexually abused (cOR = 4.23 and aOR = 12.25 for non-LBC).

We also observed that victims of bullying who had siblings (cOR = 2.89 and aOR = 3.85 for LBC, cOR = 2.31and aOR = 2.58 for non-LBC), or lived in rural areas (cOR = 2.66 and a OR = 2.97 for LBC, cOR = 2.60 and aOR = 2.92 for non-LBC) were more likely to suffer CSA.

## DISCUSSION

We investigated the association between bullying victimization and CSA among LBC and non-LBC from Anhui and Sichuan provinces, which was unexplored in Chinese culture
**Table 3  Adjusted associations between bullying victimization and CSA, stratified by gender, age, only child, home place, family structure.**

| | Total OR (95% CI, *p* value) | LBC OR (95% CI, *p* value) | Non-LBC OR (95% CI, *p* value) |
|---|---|---|---|
| Bullying victimization[a] | 2.35(1.68–3.30, <0.001) | 2.52(1.34–4.73,0.004) | 2.35(1.58,3.53, <0.001) |
| Gender[a] | | | |
|   Boys | 2.02(1.34–3.03,0.001) | 1.32(0.58–2.98,0.501) | 2.34(1.44–3.79,0.001) |
|   Girls | 3.40(1.81–6.38, <0.001) | 7.36(2.16–24.99,0.001) | 2.38(1.08–5.27,0.032) |
| Age (years)[a] | | | |
|   11–15 | 3.16(1.89–5.30, <0.001) | 3.42(1.18–9.93,0.023) | 3.32(1.80–6.15, <0.001) |
|   16–18 | 1.78(1.10–2.87,0.018) | 2.06(0.86–4.91,0.105) | 1.65(0.92–2.97,0.093) |
| Only child[a] | | | |
|   Yes | 2.12(1.31–3.43,0.002) | 2.33(0.95–5.67,0.063) | 2.17(1.19–3.96,0.011) |
|   No | 2.81(1.72–4.57, <0.001) | 3.85(1.37–10.85,0.011) | 2.58(1.46–4.54,0.001) |
| Home place[a] | | | |
|   Urban | 1.55(0.85–2.82,0.157) | 0.96(0.23–4.06,0.956) | 1.65(0.83–3.30,0.153) |
|   Rural | 2.82(1.86–4.26, <0.001) | 2.97(1.42–6.22,0.004) | 2.92(1.74–4.89, <0.001) |
| Family structure[a] | | | |
|   Traditional | 2.20(1.54–3.14, <0.001) | 2.69(1.35–5.34,0.005) | 2.13(1.40–3.26, <0.001) |
|   Non–traditional | 5.95(1.46–24.21,0.013) | 11.72(0.36–379.20,0.165) | 12.25(1.51–99.63,0.019) |

Notes.

[a] Adjusted for potential confounders, including age, gender, only child, home place, family structure, relationship with mother, relationship with father, parental educational level.

before. Our results indicated that CSA was a major concern and affected a significant proportion of LBC, especially in those who were more vulnerable, namely victims of bullying.

The prevalence of CSA in the present study was 22.89% in LBC and 19.17% in non-LBC, which echoes the findings of other studies abroad ranging from 7.00% to 29.87% (*Afifi et al., 2003*; *Madu, Idemudia & Jegede, 2001*). The prevalence was the highest(29.87%) in high school students (*SAJ & Idemudia, 2001*) and lowest (7.00%) in primary school students (*Afifi et al., 2003*). Our results were within the range because our participants included both junior and senior high students.

Few existing studies have reported the dose–response relationship between bullying victimization and CSA. One review article suggested that exposure to bullying was an identified correlate of sexual abuse; however, the authors didn't mention the contribution of bullying victimization to the development of sexual abuse (*Meinck et al., 2015b*).The literature on poly-victimization claimed that victims of abuse were prone to suffer other types of abuses (*Finkelhor, Ormrod & Turner, 2007*), which was confirmed in our study where we observed a potential dose—response relationship between bullying victimization and CSA, the high score victims were accompanied by higher risk of being sexually abused in both LBC (aOR = 2.52, 95% CI [1.34–4.73]) and non-LBC (aOR = 2.35, 95% CI [1.58–3.53]).

Based on existing research (*Wolke et al., 2013*), we identified and controlled a range of potential confounders (*Andersson & Ho-Foster, 2008*; *Austin, Shanahan & Zvara,*

*2018*; *Carey et al., 2008*; *Ibrahim et al., 2008*; *Shams, Garmaroudi & Nedjat, 2017*; *White & Warner, 2015*). After adjusting for potential confounders, the contribution of bullying victimization to the sexual abuse increased from 6.01 times (95% CI [1.90–19.09]) to 7.36 times (95% CI [2.16–24.99]) among left-behind girls.

We noticed that the association between bullying victimization and CAS was significant only among the female LBC (aOR = 7.36,95% CI [2.16–24.99]), suggesting that female victims are more likely to be sexually abused (*Meinck, Cluver & Boyes, 2015a*). Besides, CSA was associated with bullying victimization among LBC aged 11–15 years (aOR = 3.42, 95% CI [1.18–9.93]). However, the association wasn't significant within the older group, which indicated early adolescence was a critical transitional period (11–15 years old) for sexual abuse, which was worthy of public attention (*Hamil-Luker, Land & Blau, 2004*). Thus, CSA caused by bullying victimization is preventable, if the school or main caregivers could put more attention on this age group.

The risk of victims of bullying suffered sexual abuse was 3.85 times in LBC who had siblings, while the risk was not observed in LBC who had no siblings. The possible explanation behind was LBC who had siblings got less care from parents than their only-child peers did, and parental care could prevent victims of bullying from being sexually abused (*Dai et al., 2017*). Victims of bullying living with or without traditional family structure were more likely to suffer sexual abuse (for traditional family: aOR = 2.20, 95%CI [1.54–3.14]; for non-traditional family: aOR = 5.95, 95% CI [1.46–24.21]). Parents divorced or died may have a worse contribution to CSA in non-LBC, while living apart could threaten the relationship and secure attachment with parents in LBC (*Wang et al., 2015*). Parental departure actually could reduce parents' support and supervision and increase the risk of suffering abuse and neglect (*Zhao, Liu & Wang, 2015*). One study indicated that the care of mother was the key protective factor for the mental resilience of LBC, and the probable cause was that most children felt the strongest attachment to their mothers, given that in most cases mother was the primary caregiver and was difficult to be replaced (*Qiaolan et al., 2011*). What's more, other studies have shown that parental psychiatric problems, domestic violence, and disinterested mother could contribute to CSA (*Afifi et al., 2003*).

Bullying victimization was associated with CSA in both LBC (aOR = 2.97, 95%CI [1.42–6.22]) and non-LBC (aOR = 2.92, 95%CI [1.74–4.89]) who lived in rural areas. Rural and urban areas differed from some important variables including socioeconomic status, liberalism/conservatism, and poverty rates (*Kowalski, Giumetti & Limber, 2017*). Compared with their urban counterparts, rural children might get less care from their parents when they were bullied in school because of the poor living environment, which increased the risk of being sexually abused (*Xia et al., 2010*).

## Limitations

Our study has some limitations. First, the study is a cross-sectional design, which hinders its ability to infer causality. The relationship between bullying victimization and CSA may be bidirectional (*Duncan, 1999*; *Meinck, Cluver & Boyes, 2015a*), and further studies with longitudinal design are needed. Second, the self-reported experience of

bullying victimization and CSA may result in recall bias. Thus, the prevalence of bullying victimization and CSA could be underestimated. However, the CASI technology we adopted in the survey can protect the privacy of children better and substantially reduce the information bias (*Brown, Vanable & Eriksen, 2008*). Third, our findings from school students may not be generalized to adolescents that have dropped out of school. Fourth, we simplify condensed the experience of CSA with ten items in one dummy variable (yes/no), which may lead to biased results of the current study. The effect of bullying victimization on severity of CSA need to be explored in future studies.

Despite these limitations, our findings demonstrate that bullying victimization increases the risk of CSA and add evidence that individual characteristics, including younger, female, only-child, rural area, and non-traditional family, may amplify the adverse effects of being bullied on CSA among Chinese LBC. Our study suggests that anti-bullying interventions targeted at those vulnerable populations may enhance the protective effects on CSA among over 61.02 million Chinese LBC.

## CONCLUSION

Our findings attest that victims of bullying among both LBC and non-LBC are more likely to suffer CSA, which has significant implications for bullying intervention. Screening for peer bullying in schools may help reduce the risk of CSA, given that CSA, in most cases, is insidious and subtle. Our results highlight the female victims of bullying are more vulnerable to CSA. Besides, the children who live in rural areas or who have siblings are also susceptible to CSA after being bullied. Thus, more attention should be paid to these vulnerable children to lower the risk of CSA.

Since the present study is the first step in understanding the relationship between bullying victimization and CSA among Chinese LBC, future studies should also point to the contribution of social support to resilience enhancement in victims of school bullying.

## ACKNOWLEDGEMENTS

The authors thank site coordinators for their hard work. The authors also appreciate the cooperation of the children involved.

### Funding

This study was funded by scientific research program of the Shanghai Municipal Health and Family Planning Commission (No. 201540091), the Innovation-oriented Science and Technology Grant from NPFPC Key Laboratory of Reproduction Regulation (No.CX2017-5) and the United Nations Population Fund (Grant No.PL-SSA20). The funders had no role in study design, data collection and analysis, decision to publish, or preparation of the manuscript.

## Grant Disclosures

The following grant information was disclosed by the authors:

Shanghai Municipal Health and Family Planning Commission: 201540091.

NPFPC Key Laboratory of Reproduction Regulation: CX2017-5.

United Nations Population Fund: PL-SSA20.

## Competing Interests

The authors declare there are no competing interests.

## Author Contributions

- Li Yan conceived and designed the experiments, analyzed the data, contributed reagents/materials/analysis tools, prepared figures and/or tables, authored or reviewed drafts of the paper.
- Qianqian Zhu performed the experiments, contributed reagents/materials/analysis tools, authored or reviewed drafts of the paper.
- Xiaowen Tu and Chaohua Lou contributed reagents/materials/analysis tools, authored or reviewed drafts of the paper.
- Xiayun Zuo and Chunyan Yu performed the experiments, authored or reviewed drafts of the paper.
- Qiguo Lian conceived and designed the experiments, performed the experiments, analyzed the data, prepared figures and/or tables, authored or reviewed drafts of the paper, approved the final draft.

## Human Ethics

The following information was supplied relating to ethical approvals (i.e., approving body and any reference numbers):

The study was reviewed and approved by the institutional review board of Shanghai Institute of Planned Parenthood Research (PJ2015-5).

## Data Availability

The raw data are provided in a Supplemental File.

## Supplemental Information

Supplemental information for this article can be found online at http://dx.doi.org/10.7717/peerj.4865#supplemental-information.

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
