# Peer review of "Bullying victimization and child sexual abuse among left-behind and non-left-behind children in China"

_PeerJ, doi:10.7717/peerj.4865_

## Round 0.1 · original submission · Major Revisions

Dear authors,

Your paper has been carefully read by two external reviewers and me, and we consider that it would have scientific merit when some changes are applied to the text. Consequently, your manuscript needs to be re-evaluated (MAJOR REVISION).

With respect and warm regards,
Dr Palazón-Bru (academic editor for PeerJ)

Reviewer 1 ·

Basic reporting

1. Minor grammatical errors and misuse of punctuation and prepositions (e.g. line 91, 125); need to proofread.

2. Unclear writing/irrelevant content here and there:
- line 31, “bullying victimization was higher risk factor for CSA”. Higher compared with what?
- line 34, "CSA was higher among school girls," but the supporting numbers are for "left-behind CHILDREN vs. non-left-behind children", should be "left-behind GIRLS vs. non-left-behind girls"
- line 82, "the link.. remains to be explored". Do you mean "unexplored"?
- line 90, target population aged 11-18 from middle schools", should be "middle school and high school"?
- line 125, assessed by 10 items, but only listed 5 items?
- line 130, does "home place" mean the child's current place of residence? or place of household registration?
- line 186: "direct and indirect effects of CSA": what are they? Which ones are direct? Which are indirect?
- line 216, unclear meaning of "a rebellion stage" and how it relates to your discussion.
- line 219, "children having good relationship with family are less depressed", seem irrelevant as depression was not mentioned anywhere in your lit review/study.
- line 247, the measurement for "social economic level" was discussed as a limitation but was not part of your measure/study.

3. Literature review needs more context, details of existing studies, and re-organization.
- line 60, what year was this national prevalence for? Are the numbers up to date?
- line 69, "left-behind children, a special group in rural China..." How are they special? Special in terms of what? Need a little more context here.
- line 79, you cited Meinck's study, what was their target population? What was the strength of prediction? Need more details.
- some parts in discussion should be moved to lit review: line 192-203.
- need more literature to support why you choose the covariates.

4. Citations should use last names only.

5. The table 1 formatting makes it unclear between which groups the difference was significant.

6. Titles for table 2 & 3 are misplaced.

Experimental design

1. research questions and hypotheses are unclear.

2. rationale for the study: your identified research gap is “the link between CSA and bullying victimization remains to be unexplored” (line 81), but you cited studies that already explored this link (line 53 & 67)? Perhaps highlight your contribution is no existing studies looked at Chinese left-behind children.

3. method: specify how did you recruit participants, how to select sampled schools, what are the school characteristics, and which grades are the students from.

4. ambiguous explanation for dropped cases: is it because missing data (line 96) or age over 18 (line 155)?

5. ethical concern: need to provide child oral assent form/template

6. did any parents/guardians refuse to participate, since their child may be sexually abused by them?

7. was the survey translated into Chinese? Were the questionnaires culturally applicable?

8. how was the survey administered? By whom? You mentioned it’s done in computer labs, so was it a group setting? Would that bias your results if students were uncomfortable to report when their peers were in the same room? Were anyone on site to answer questions during the survey? Were severely-abused/bullied cases referred to social worker/counselor?

9. unclear measurement: “parental educational level”, of father or mother?

10. data analysis: need to report relative change between crude and adjusted estimates in order to assess effects of confounding variables. Right now you only discussed table 3 results, table 2 was not discussed at all.

11. data analysis: since your main IV is bullying victimization while controlling for all other covariates, why not report multivariate regression tables?

Validity of the findings

1. If you did not measure the time point of bullying/sexual abuse, then you cannot establish causal inference (as briefly noted in your limitation section). Need to avoid using causal language.

2. Some findings cannot be supported by your study:
- line 204, “the worse the degree of being bullied, the higher risk of sexual abuse” –you did not test the degree of bullying in analysis, only being bullied or not.
- line 238: you did not measure marital status in your analysis.
- line 247: socioeconomic status, you did not mention in your tables/measures.

3. Discussion can be more focused on your main IV (bullying victimization), rather than the covariates (gender, age…)

4. potential selection bias: your sample was from middle school and high school, would younger left-behind children and those dropped out of school be more vulnerable? Peers and being in school are protective factors. In other words, your sample may be in better status than other left-behind children. Maybe add as a limitation.

5. what does your findings mean? What policy/practice implications do they bring?

Additional comments

This study discussed an important issue for a large high-risk population in China: left-behind children. It potentially could make contribution to literature, but does require more introduction of the context and existing literature, clearer research question, more rigorous reporting in methodology and results, and elaboration on findings and limitations.

Reviewer 2 ·

Basic reporting

1. It seems there are few studies focused on CSA among LBC and this paper is likely to fill this gap of knowledge in some way. However, it is better if the authors could provide more review on studies on factors that are related to CSA and bullying victimization, justifying the inclusion of covariates in the current study.
2. There is a lack of theoretical review in this paper and theoretical framework is missing. The authors should also justify the theoretical logic under this study.
3. Please clarify the statistics used at line 45, are they description for China or for the whole world?
4. Languages should be improved or re-edited in some places. For example, at line 56-57, “victims of bullies” and “bully-victims” are reiterated for the same meaning. The description of findings in a previous study at line 67-68 is better to be further clarified or restated. The meaning is not clear.
5. The formats of in-text citations and references are not correct in some parts of this paper. For example, at line 199, should be “Duncan (1999) reported that…..”, while should remove (Duncan, 1999) at line 2000. Same thing for Mansbach at line 200. The authors should check these kinds of in-text citations over the paper. In addition, many papers were not referred correctly in style in the section of references, including but not limited to line 275, 283,295,305,317,319,327, 330 etc.

Experimental design

1. Research questions may better to be clearly defined although the knowledge gap has been identified in the introduction section.
2. The research design has been illustrated in detail. However, regarding measures of variables, it is quite confused for measurements of child sexual abuse. The authors may need to further clarify it. At line 125-126, the authors mentioned 10 items but only listed five here, or it should be five aspects and each was measured by 2 forms, thus in total 10 items? In terms of covariates, it seems in the later part of this paper (see line 150,163, 223 and 241) the authors mentioned parents’ marital status but no relevant information was shown in the section of Covariates (line 129). Nor in the tables. The authors should explain for that.
3. It seems the statistical analysis described does not match what was shown in Table 3 and results at lines 175-183. The results at lines 175-183 seem to present results for binary logistic regressions for the total sample, LBC and non-LBC, respectively, controlling for covariates. However, what has been shown in Table 3 and described in statistical analysis (see line 150-151, e.g. subgroup analysis) seem to be conducting analysis for each group of gender, age, home place, parent marriage etc. and investigate the associations between bullying victimization and CSA. The authors need to be clear of that.

Validity of the findings

1. Despite the confusing presentation of Table 2 and 3, the results written since line 152-183 seem to show the associations between bullying victimization and CSA, as well as its difference for LBC and non-LBC. In addition, other associations for covariates were also illustrated. As mentioned above, the results of parental marriage cannot be obtained from the current table and where are they from? Also, pay attention to it in the discussion part.
2. The authors explained most of the findings connecting with existing literature. However, what is the theoretical contribution of this study?
3. Limitations were not identified enough. Particularly, the authors simplify the variable of CSA as a dummy variable, for which it is coded as one if any of those items were answered with yes. CSA with the five aspects at line 126 may hold different levels of severity from each other. Simply condensing information in one dummy variable may lead to biased results of the current study. The authors should identify and explain the potential biases of results in the current study. I would also suggest trying multinomial logistic regression.

Additional comments

This paper focuses on an interesting topic that is related to bullying victimization and child sexual abuse, especially contributing to relevant knowledge for left-behind children in China. For most parts of this paper, it is well written in English. However, the current study seems not sound enough and there are some points suggested being improved and clarified.

---

## Round 0.2 · accepted · Accept

Dear authors,

I have evaluated your revision and in addition Reviewer 1 has confirmed that is it now Acceptable. As a result, it is a pleasure to inform you that your paper has been accepted for publication in its current form in PeerJ.

Congratulations!

With respect and warm regards,
Dr Palazón-Bru (academic editor for PeerJ)

Reviewer 2 ·

Basic reporting

No comment

Experimental design

No comment

Validity of the findings

no comment

Additional comments

No comment